# Comparison on Reduction of VOCs Emissions from Radiata Pine (*Pinus Radiata D.* Don) between Sodium Bicarbonate and Ozone Treatments

**DOI:** 10.3390/molecules25030471

**Published:** 2020-01-22

**Authors:** Ye Qin, Fei Qi, Zhiping Wang, Xianbao Cheng, Botao Li, Anmin Huang, Ru Liu

**Affiliations:** 1Research Institute of Wood Industry, Chinese Academy of Forestry, Beijing 100091, China; qinye@caf.ac.cn (Y.Q.); 15321593606@163.com (Z.W.); cheng.xianbao@163.com (X.C.); 18360736480@163.com (B.L.); liuru@criwi.org.cn (R.L.); 2Beijing Key Lab for Source Control Technology of Water Pollution, College of Environmental Science and Engineering, Beijing Forestry University, Beijing 100083, China; qifei@bjfu.edu.cn

**Keywords:** indoor air quality, volatile organic compounds, sodium bicarbonate treatment, ozone treatment, radiata pine

## Abstract

Volatile organic compounds (VOCs) in wood furniture are an important factor that affects indoor air quality. In this study, radiata pine (*Pinus radiata D.* Don) was treated with sodium bicarbonate and ozone aqueous solution to reduce the VOC contents without sacrificing mechanical properties. The VOCs of radiata pine were identified by gas chromatography-mass spectrometry (GC-MS), and the functional group changes of wood samples were characterized by Fourier-transform infrared spectroscopy (FTIR). The results showed that the main VOCs of radiata pine include alkenes, aldehydes, and esters. The sodium bicarbonate and ozone treatments almost eliminated the VOC contents of radiata pine. The two treatments mentioned above had little effect on compressive strength and surface color of radiata pine.

## 1. Introduction

With the expansion of requirements for living quality, people have paid increasingly more attention to the impact of volatile organic compounds (VOCs) released from furniture, which is an important factor that affects indoor air quality. As one of the most common materials for furniture, the release of VOC in wood and wood-based composites has attracted significant attention from researchers [1]. High concentrations of the total content of VOC (TVOC) are highly toxic to the human body [2]; many compounds in VOCs have carcinogenic risks [3]. VOCs in wood include alkenes, alcohols, aldehydes, and ketones [4], and the main VOC constituents of radiata pine are composed of α-pinenes and β-pinenes. Most children’s furniture in China is made of radiata pine; therefore, the reduction of the VOC content of radiata pine has great significance.

Heat treatment can effectively remove aldehydes from wood [5], and is an effective wood processing technology that affects the physical and chemical properties of wood [6]. Hyttinen et al. [7] reported that heat treatment can significantly reduce the emission of VOCs, such as terpenes and aldehydes, from European poplar samples due to the degradation or evaporation of small molecules. Additionally, Jiang et al. [8] stated that caproaldehyde and pentaldehyde in wood VOCs were highly sensitive to temperature. In a preliminary study, it was found that the main VOC content of southern yellow pine samples decreased the most when heat treatment was conducted at 220 °C [9]. However, the energy consumption of heat treatment in the wood industry is high, and it can cause other drawbacks including the change of surface color and decrease of mechanical strength [10]. As another effective method, extraction is often used to modify wood chemical properties, and affects wood thermal stability [11], cementing [12,13], and antibacterial properties [14]. Studies have shown that wood VOCs mainly originate from their extraction [15]. Therefore, the compounds in wood can be finely extracted by solvent extraction [16]. Researchers [17] have used ethanol-toluene to reduce wood VOCs. However, the content of the solvent that remains in the treated samples is high. Some studies have also shown that extraction with ethanol-cyclohexane can significantly reduce the emission of terpenes and aldehydes in pine [18]. The reduction of VOCs by extraction has not been widely used due to the high content of solvent that remains in the treated wood. It is also difficult for the extraction of large samples in the actual production process, and for the treatment of waste liquid containing chemical solvents generated by extraction.

As the VOCs of wood mainly result from extraction, easily realized chemical methods were used to treat radiata pine in this study. Alkali treatment is a common chemical treatment, and studies have shown that alkaline solutions can promote the hydrolysis of fats and resins in wood [19]. It has been found that the sodium hydroxide solution can partially remove terpenes and aldehydes from pine extract [18]. Sodium bicarbonate has been used as a buffer to significantly reduce the VOC contents in styrene-acrylate emulsion [20] and acrylic elastic emulsion [21]. Additionally, researchers have found that treating cannabis leaves with sodium bicarbonate could inhibit their VOC acidity and reduce aldehyde emissions [22]. However, the treatment of wood with sodium bicarbonate will lead to the increase of hydroxyl on the surface of wood [23], which can easily decompose hemicellulose and lignin, thereby damaging the structure of wood [24] and affecting its use. In this experiment, weakly alkaline sodium bicarbonate was used to treat the radiata pine for reducing VOCs, and exhibited less impact on the performance of samples while removing VOCs. In addition, oxidants can be used to simply and efficiently treat wood extracts and remove VOCs. Studies have shown that removing VOCs from pine wood with hydrogen peroxide as an oxidant can reduce the emission of monoterpenes [25]. As a type of oxidant, ozone can effectively reduce the TVOC concentration of sewage [26] and promote the photodegradation of VOCs in the atmosphere [27]. Researchers have found that wood reacts with an ozone solution [28], which degrades partial extracts in the wood [29]. Therefore, ozone treatment can also play a positive role in the removal of VOCs from wood and creates fewer by-products [30].

As mentioned previously, VOCs of wooden furniture are important factors that affect indoor air quality [31,32], which in turn affects human health. Currently, there are few technologies to remove wood VOCs without sacrificed other properties of wood. Therefore, in this paper, radiata pine was respectively treated by sodium bicarbonate and ozone aqueous solution to reduce its VOC content. Solid-phase microextraction (SPME) coupled with gas chromatography-mass spectrometry (GC-MS) was used to determine the changes in VOC content [33,34]. Fourier-transform infrared spectroscopy (FTIR) was used to analyze the changes in functional groups of various VOC components [35,36]. The method proposed in this paper can effectively reduce the VOC content of radiata pine, and promote the use of radiata pine furniture due to its decreased VOCs release.

## 2. Results

### 2.1. FTIR Results and Analysis of Functional Groups

The FTIR spectra of the untreated and treated radiata pine samples are presented in Figure 1. The characteristic chemical band of wood in the FTIR spectra was determined according to the existing literature [37]. The detailed peak positions are listed in Table 1.

From the FTIR results, some chemical changes were found to have occurred after the treatments with different processing methods. As shown in Table 2, the relatively stable aromatic ether at 1270 cm^−1^ was selected as the internal standard control peak and the control values of other peak areas were obtained.

The absorption peaks at 1640 cm^−1^, 1510 cm^−1^, and 1425 cm^−1^ were attributable to the vibration of the aromatic ring framework. The absorption peak strengths of the treated samples decreased compared to those of the untreated samples. Both of these treatments will cause C=C to be destroyed. Alkali treatment can remove lignin in wood powder, exposed more cellulose and increase the concentration of hydroxyl on the surface of cellulose [38]. Ozone treatment can oxidize lignin and lead to hemicellulose degradation. The absorption peak at 1730 cm^−1^ is attributable to C=O. The absorption peaks of the treated radiata pine samples were slightly moderate here, indicating that the sodium bicarbonate and ozone treatments reduced the amount of aldehyde compounds. The wavelength at 1640 cm^−1^ is the wide absorption peak of C=C stretching vibration, and the strength of this characteristic peak of the ozone-treated samples was slightly lower than that of the untreated samples, indicating that C=C was fractured under the action of ozone. The absorption peaks at 1160 cm^−1^ and 1109 cm^−1^ were generated by expansion vibration absorption of aliphatic ether. Table 2 shows that the relative value of fat ether of ozone-treated samples is smaller than that of sodium bicarbonate treated samples. Both of these reactions were accompanied by the formation of small molecules, so there is an increase in the group at peaks 1160 and 1109.

### 2.2. GC-MS

The occurrence times, specific names, and relative contents of different VOC compounds in radiata pine before and after treatment as determined by GC-MS are exhibited in Table 3. The corresponding characteristic peaks in the GC-MS chromatograms of the samples are presented in Figure 2.

Table 3 lists the main VOCs before and after treatment. The high-content (>5%) compounds in the untreated radiata pine included Methoxyacetic acid, pentyl ester (peak 2, 9.29%), Furfural (peak 6, 19.98%), 3,5-Dimethylpyrazole (peak 9, 11.19%), 1H-3a,7-Methanoazulene, 2,3,4,7,8,8a-hexahydro-3,6,8,8-tetramethyl-, [3R-(3.alpha.,3a.beta.,7.beta.,8a.alpha.)]- (peak 25, 5.28%)—and Di-epi-.alpha.-cedrene—(peak 28, 38.63%). After the sodium bicarbonate and ozone treatments, the contents of these compounds changed greatly. Both treatments effectively reduced the VOC contents, and these major compounds were significantly reduced. In general, the VOC contents of the sodium bicarbonate-treated samples were lower than those of the ozone-treated samples.

However, these two treatments had different effects on the contents of different VOCs. The sodium bicarbonate treatment reduced the contents of 1H-3a,7-Methanoazulene, 2,3,4,7,8,8a-hexahydro-3,6,8,8-tetramethyl-, and [3R-(3.alpha.,3a.beta.,7.beta.,8a.alpha.)]—(peak 25)—more efficiently. The ozone treatment more efficiently reduced the contents of other compounds, including Hexanal (peak 5), Benzaldehyde (peak 12), Benzyl alcohol (peak 20), 1H-3a,7-Methanoazulene, octahydro-3,8,8-trimethyl-6-methylene-, [3R-(3.alpha.,3a.beta.,7.beta.,8a.alpha.)]—(peak 26)—Di-epi-.alpha.-cedrene—(peak 28)—Silane, and [[4-[1,2-bis[(trimethylsilyl)oxy]ethyl]-1,2-phenylene]bis(oxy)]bis[trimethyl—(peak 29).

All compounds within the radiata pine VOC range were classified (Table 4) as alkenes, aldehydes, esters, alkanes, alcohols, ketones, amines, ethers, and others. Figure 3 indicates that, after the sodium bicarbonate and ozone treatments, the VOC contents of main compounds such as alkenes, aldehydes, and esters in the radiata pine samples were significantly reduced.

Studies had found that pinene compounds in pine were easy to be oxidized, which were usually shown as aromatization [39]. Terpenes reacted quickly with ozone or hydroxyl radicals to produce different organic compounds, including aldehydes and small amounts of ketones [40].

Hoigne [41] initially proposed the advanced oxidation technology and mechanism, which is a process of producing free radicals by different methods. The oxidation of organic compounds by ·OH radicals can be divided into three reactions [42], and the reaction formula is as follows.
RH + OH → H2O + R

It can be seen that the ·OH radical can stimulate the inert hydrogen on the organic ring and react with dehydrogenation to form an R radical, which becomes the initiator of further oxidation. For the electrophilic addition reaction, the reaction formula is as follows.
OH + PHX → HOPHX

Finally, for the electron transfer reaction, the reaction formula is as follows.
OH + RX → RX^+^ + OH

Ozone interacted with water molecules to produce ·OH radicals and superoxide radicals to act on samples [43]. Studies have found that lignin degrades in ozone [44], cellulose glycoside bonds break [29], and free radicals can enter the internal reactions of the samples. Studies on ozone have shown that when ozone passes into alkenes-containing solutions, carboxylic acids are generated under oxidation conditions [45]. It can be seen that the VOC compounds in the untreated samples were mainly composed of furfural, and some furfural may degenerate into polysaccharide under the action of thermal stress [46]. Part of the alcohol can react with sodium bicarbonate to form sodium alcohols (potassium alcohols) and water.

Most ester groups in the wood come from the acetyl groups in hemicellulose components. In alkaline solutions, ester groups are hydrolyzed and free hydroxyl groups are released, leaving acetate ions in the solution [47]. The reaction formula is as follows.

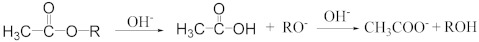


Researchers have found that when cannabis leaves are treated with sodium bicarbonate, the aldehyde contents are reduced compared with untreated leaves [22]. In a reaction study of ozone and aldehydes, it was found that the slow reaction of ozone and aldehydes occurs in the atmosphere [48]. In this experiment, water was used as the reaction medium, and the long-term exposure of a high concentration of ozone had a significant impact on the aldehydes in the sample, resulting in a decline in their contents. The sodium bicarbonate and ozone treatments presented a good removal ability of the VOC ester compounds; sodium bicarbonate can promote the hydrolysis of esters [23,24], and it has been found that ozone has a good effect on esters in the treatment of municipal sewage [49].

The ozone and sodium bicarbonate treated the ethers in the VOC of the samples, which were the products of the replacement of hydrogen in the hydroxyl groups of alcohol by hydrocarbon groups. The VOC in the untreated samples was mainly acetone, which was easily soluble in water and oxidized by strong oxidants. The amine content increased after the ozone treatment, mainly producing actinobolin and n-Hexylmethylamine; the sodium bicarbonate treatment completely removed amines.

### 2.3. Analysis of the Effects of Treatments on Physical and Microscopic Properties

#### 2.3.1. Strength Properties and pH Values of Samples

The compressive strength and deformation of each sample before and after treatment were tested by the universal testing machine, and the results are exhibited in Table 5.

Table 5 shows that the compressive strength of the samples treated with sodium bicarbonate decreased by 12.12%, and that of the samples treated with ozone decreased by 6.55%. It has been reported that the compressive strength of wood samples decreased by about 5% when the heat treatment temperature was 110 °C [50]. With the increase of temperature, the compressive strength of wood samples decreased gradually. When the temperature reached 200 °C, the compressive strength of the samples decreased by about 35% after 10 h of heating [51]. In summary, compared with the reduction of wood VOCs by heat treatment, the two proposed treatment methods had a small impact on the compressive strength of wood.

It can also be seen from Table 5 that the acidity of the sample treated with sodium bicarbonate was decreased compared with that of the untreated radiata pine, indicating that the sodium bicarbonate treatment changed the acidity and alkalinity of the samples, while the ozone treatment did not have any effect.

#### 2.3.2. Surface Colors and Micro-Textures of Samples

As shown in Figure 4, the difference in surface color of the samples treated by the two treatment methods was small compared with the untreated samples. The ozone-treated samples became lighter, and the sodium bicarbonate-treated samples became darker.

As shown in Figure 5, the SEM images of the radiata pine samples illustrate clear features, such as fibers and pores. The pore morphology of the untreated samples was complete and slightly protruding, and the fiber morphology was complete. The SEM images of samples treated with sodium bicarbonate reveal that some striations shrank inward and some changes took place in the fiber, indicating that the internal structure of the sample changed under the action of the sodium bicarbonate; the crystallinity of cellulose decreased, and the binding layers of lignin or hemicellulose and cellulose were destroyed [52]. The compressive strength of the samples treated with sodium bicarbonate decreased.

The SEM images of radiata pine samples after ozone treatment reveal that the striated pores were expanded and broken in the middle. Ozone affected the double bonds in the VOCs of radiata pine samples, and also had a bleaching effect [53,54]. The fiber structure of the radiata pine was changed, which may have been caused by the ozone action of cellulose; thus, the compressive strength of the samples after ozone treatment was slightly reduced [29].

## 3. Materials and Methods

### 3.1. Materials

Radiata pine (*Pinus radiata D*. Don) was imported from New Zealand by Linyi Fuhe Wood Co., Ltd., Shandong, China. The initial moisture content of the test materials was between 8% and 10%, and the materials were stored at 10–40 °C with a relative humidity of 30–70% for one year.

### 3.2. Preparation of Wood Samples

The radiata pine logs were processed into 20 × 20 × 30 mm^3^ experimental samples. They were dried in an oven (Shanghai Yihuan Experimental Instrument Co., Ltd., Shanghai, China) at 103 °C until a constant weight was reached. Experimental samples were divided into three groups. One was the control group, and the other two were the experimental groups.

### 3.3. Treatment

#### 3.3.1. Sodium Bicarbonate Treatment

Sodium bicarbonate solution with a mass fraction of 1% was used for the impregnation of wood specimens. The samples were placed in a vacuum drying chamber (Shanghai Boxun Industry and Commerce Co., Ltd., Shanghai, China). They were first vacuumed to −0.1 MPa, and maintained at a temperature of 80 °C for 60 min. After maceration, samples were placed in a beaker that contained an appropriate amount of distilled water to dilute the mass fraction to 0.25%. The beaker was then placed on an electric stove (Tianjin Zhonghuan Science and Technology Development Company, Tianjin, China) and heated for 60 min. After, the wood samples were removed and air-dried for a few days before drying them in an oven.

#### 3.3.2. Ozone Treatment

An ozone generator was obtained from Beijing Linke Science and Technology Co., Ltd., Beijing, China. It was combined with a reaction vessel to form the experimental device. The velocity of the ozone was 250 mL·min^−1^. The wood samples were placed in the generator and treated for 60 min.

### 3.4. Fourier-Transform Infrared Spectroscopy

FTIR was used to analyze the chemical changes of the wood samples at room temperature (Nikolai Instruments, WI, USA). Potassium bromide (KBr) was used as the background spectrum. Before spectral collection, the dried sample powder was mixed with KBr at a weight ratio of 1:100. Each sample was scanned 10 times in a spectral range from 400 cm^−1^ to 4000 cm^−1^ with a resolution of 0.4 cm^−1^.

### 3.5. Gas Chromatography-Mass Spectrometry

#### 3.5.1. Solid Phase Micro Extraction (SPME)

The wood powder was placed in a sealed overhead bottle, and placed in a 60 °C water bath pot after using a 100 μm PDMS SPME head to adsorb the volatile organic compounds. The adsorption time was 40 min. The solid phase micro-extraction head was then inserted into the gas chromatographic inlet, and analysis was conducted for 4 min.

#### 3.5.2. Chromatographic Conditions

Helium gas was used as the carrier gas at a constant flow rate of 1.0 mL/min and an injection port temperature of 250 °C. The initial temperature was 40 °C, which was then increased at a rate of 5 °C/min to 230 °C, and finally maintained for 5 min.

#### 3.5.3. Mass Spectrometry

The analysis conditions included a GC-MS interface temperature of 280 °C, an ion source temperature of 230 °C, and a quadrupole temperature of 150 °C. The ionization mode was electron ionization, the electron energy was 70 eV, and mass scanning ranged from 30 *m*/*z* to 550 *m*/*z*.

#### 3.5.4. Qualitative and Quantitative Analyses of Compounds

Compounds were identified by comparing the MS spectra to the NIST 2.0 library. The relative content of each chemical component was calculated by area normalization, and the average values of the three replicates were reported.

### 3.6. Contrast Correlation Properties of Samples before and after Treatment

#### 3.6.1. pH Value of Samples

The hot water extraction method was used to determine the pH values of the wood blocks (Shanghai Yidian Scientific Instrument Co., Ltd., Shanghai, China) before and after treatment.

#### 3.6.2. Strength Properties of Samples

Samples of radiata pine were treated with different methods. The untreated samples were placed in an airy atmosphere without direct exposure to the sun for one week, and were then completely dried. Each group consisted of 10 samples, amounting to a total of 30 samples in three groups. The compressive resistance of the samples in the dried condition were tested by a universal mechanical testing machine (Jinan Times Assay Instrument Co., Ltd., Shandong, China), and the changes of the toughness of the radiata pine were obtained from the time of crushing.

#### 3.6.3. Surface Colors and Micro-Morphologies of Samples

Thirty samples of radiata pine with virtually indistinguishable surface color were divided into three groups. One group was the control group, and the other two groups were treated with sodium bicarbonate and ozone, respectively. The surface colors of the radiata pine samples before and after treatment were compared.

A scanning electron microscope (Carl Zeiss, Inc., Jena, Germany) was used to observe the structures and morphologies of the samples.

## 4. Conclusions

The results demonstrated that VOCs of the untreated radiata pine samples mainly included alkenes, aldehydes, and esters. Compared with the VOCs of the untreated radiata pine, the ozone treatment of the pine samples produced trace ethers and amines. Sodium bicarbonate and ozone treatments could efficiently decrease the VOC contents of radiata pine samples. The VOCs release of the sodium bicarbonate and ozone-treated samples decreased by 88.91% and 89.20%, respectively. The sodium bicarbonate and ozone treatments reduced the compressive strength of radiata pine by 12.2% and 6.55%, respectively; these are much lower values than the compressive strength reduction value caused by heat treatment. Additionally, compared with the results of heat treatment, these two treatments resulted in slight color changes in the surfaces of the samples. Therefore, this study provided two effective VOC removal methods for radiata pine furniture production, and the results are promising for application in the industrial production process.

## Figures and Tables

**Figure 1 molecules-25-00471-f001:**
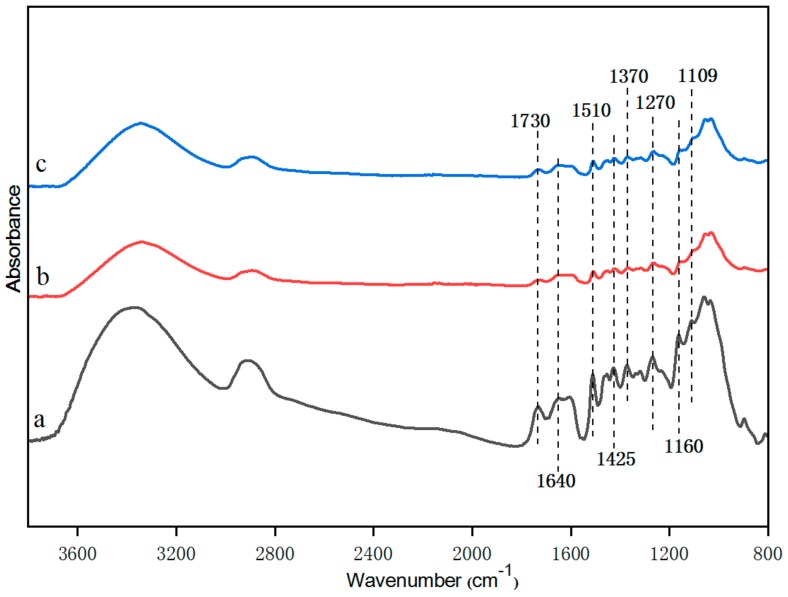
Fourier-transform infrared (FTIR) spectra: (**a**) untreated radiata pine (RP), (**b**) sodium bicarbonate-treated RP, (**c**) ozone-treated RP.

**Figure 2 molecules-25-00471-f002:**
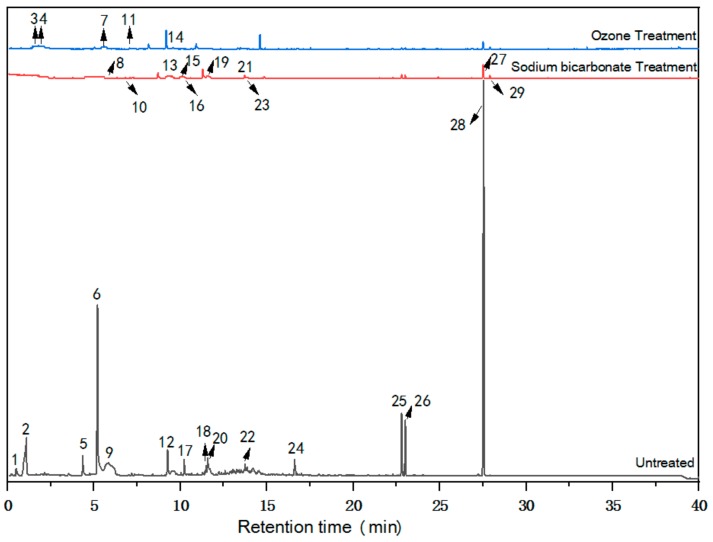
Gas chromatography-mass spectrometry (GC-MS) chromatograms of untreated and treated RP.

**Figure 3 molecules-25-00471-f003:**
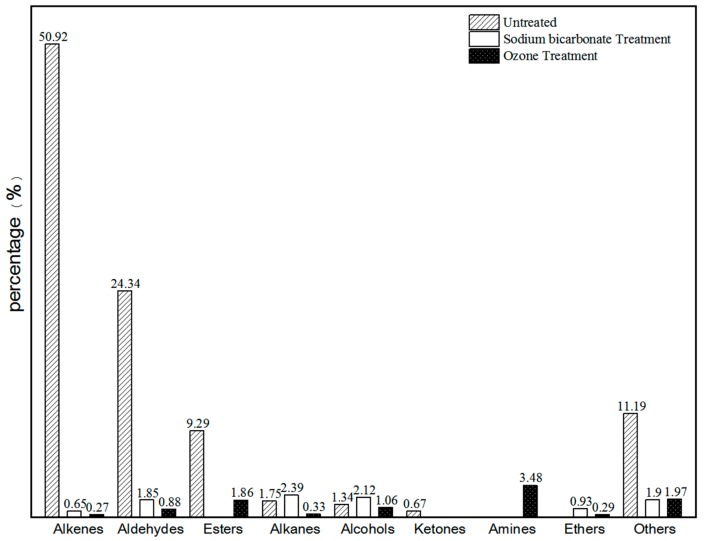
Contents of VOC groups in RP before and after different treatments.

**Figure 4 molecules-25-00471-f004:**
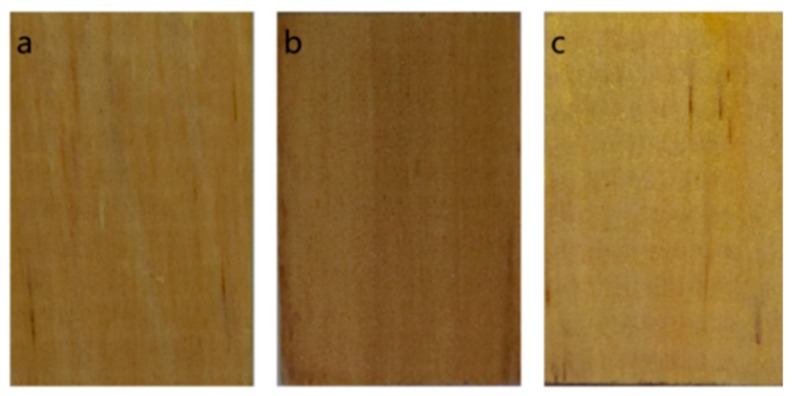
Images of surfaces of RP. (**a**) untreated sample, (**b**) sodium bicarbonate-treated sample, (**c**) ozone-treated sample.

**Figure 5 molecules-25-00471-f005:**
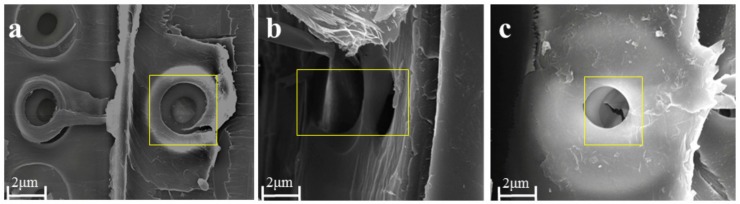
SEM images of RP: (**a**) untreated sample, (**b**) sodium bicarbonate-treated sample, (**c**) ozone-treated sample.

**Table 1 molecules-25-00471-t001:** Main bands of the infrared spectrum of RP and their functional groups.

Wave Number (cm^−1^)	Functional Groups
1730	C=O stretching of acetyl or carboxylic acid
1640	C=C stretching of the aromatic ring
1510	C=C stretching of the aromatic ring
1425	C-H plane deformation (aromatic ring skeleton)
1370	C-H bending vibration
1270	Benzene ring–oxygen bond stretching vibration (lignin)
1160	C-O-C anti-symmetric telescopic vibration
1109	C-O-C vibration

**Table 2 molecules-25-00471-t002:** The value of main infrared spectral bands of RP relative to the reference peak.

Wave Number (cm^−1^)	Relative Value		
	Untreated	Sodium Bicarbonate Treatment	Ozone Treatment
1730	1.38	0.93	1.10
1640	1.32	0.81	1.00
1510	1.49	1.30	1.16
1425	0.64	0.56	0.59
1370	0.66	0.46	0.61
1160	0.39	0.40	0.42
1109	0.28	0.49	0.26

**Table 3 molecules-25-00471-t003:** Comparison of volatile organic compound (VOC) contents of untreated and treated RP.

Peak Number	RT(min)	Compounds	Percentage (%) ^1^		
			Untreated	Sodium Bicarbonate Treatment	Ozone Treatment
1	0.50	Acetone	0.67	——	——
2	1.10	Methoxyacetic acid, pentyl ester	9.29	——	——
3	1.61	n-Hexylmethylamine	——	——	1.10
4	1.81	Actinobolin	——	——	2.38
5	4.36	Hexanal	1.85	0.20	——
6	5.21	Furfural	19.98	——	——
7	5.58	Arsenous acid, tris(trimethylsilyl) ester	——	——	1.47
8	5.83	n-Butyl ether	——	0.35	0.29
9	5.85	3,5-Dimethylpyrazole	11.19	——	——
10	7.24	1-Cyclopentyl-2,2-dimethyl-1-propanol	——	0.24	——
11	7.32	Octane, 3-methyl-	——	——	0.21
12	9.26	Benzaldehyde	2.51	1.05	0.88
13	9.35	1,3,2-Dioxaphosphorinane-2-methanol, 2-oxo-.alpha.-phenyl-	——	2.11	——
14	9.58	Butanoic acid, butyl ester	——	——	0.39
15	10.04	7H-Dibenzo[b,g]carbazole, 7-methyl-	——	0.66	1.97
16	10.20	2,5-Dihydroxyacetophenone, bis(trimethylsilyl) ether	——	0.58	——
17	10.22	Decane, 2,2-dimethyl-	1.33	——	——
18	11.49	D-Limonene	0.45	——	——
19	11.57	N-carbobenzyloxy-l-tyrosyl-l-valine	——	1.24	——
20	11.58	Benzyl alcohol	1.34	1.49	0.87
21	13.72	Nonanal	——	0.60	——
22	13.75	Undecane	0.42	——	——
23	13.84	Cyclohexanol, 4-methyl-	——	0.39	0.19
24	16.61	Cyclohexene, 1-methyl-5-(1-methylethenyl)-, (R)-	1.98	——	——
25	22.80	1H-3a,7-Methanoazulene, 2,3,4,7,8,8a-hexahydro-3,6,8,8-tetramethyl-, [3R-(3.alpha.,3a.beta.,7.beta.,8a.alpha.)]-	5.28	——	0.15
26	23.02	1H-3a,7-Methanoazulene, octahydro-3,8,8-trimethyl-6-methylene-, [3R-(3.alpha.,3a.beta.,7.beta.,8a.alpha.)]-	4.58	0.29	0.12
27	27.51	Cedrol	——	1.24	0.68
28	27.59	Di-epi-.alpha.-cedrene	38.63	0.36	——
29	27.91	Silane, [[4-[1,2-bis[(trimethylsilyl)oxy]ethyl]-1,2-phenylene]bis(oxy)]bis[trimethyl-	——	0.28	0.12
Total content			99.52	11.09	10.80

^1^ The percentage contents of the main compounds were calculated according to the peak area. ^2^ “—” indicates that the phase pair content of the substance was lower than the detection limit.

**Table 4 molecules-25-00471-t004:** Main VOCs of RP before and after treatment.

Group	Compounds
Alkenes	d-Limonene; Cyclohexene, 1-methyl-5-(1-methylethenyl)-, (R)-; 1H-3a,7-Methanoazulene, 2,3,4,7,8,8a-hexahydro-3,6,8,8-tetramethyl-, [3R-(3.alpha.,3a.beta.,7.beta.,8a.alpha.)]-; 1H-3a,7-Methanoazulene, octahydro-3,8,8-trimethyl-6-methylene-, [3R-(3.alpha.,3a.beta.,7.beta.,8a.alpha.)]-; Di-epi-.alpha.-cedrene
Aldehydes	Hexanal; Furfural; Benzaldehyde; Nonanal
Alkanes	Octane, 3-methyl-; 1,3,2-Dioxaphosphorinane-2-methanol, 2-oxo-.alpha.-phenyl-; Decane, 2,2-dimethyl-; Undecane; Silane, [[4-[1,2-bis[(trimethylsilyl)oxy]ethyl]-1,2-phenylene]bis(oxy)]bis[trimethyl-
Ketones	Acetone
Amines	n-Hexylmethylamine; Actinobolin
Esters	Methoxyacetic acid, pentyl ester; Arsenous acid, tris(trimethylsilyl) ester; Butanoic acid, butyl ester
Alcohols	1-Cyclopentyl-2,2-dimethyl-1-propanol; Benzyl alcohol; Cyclohexanol, 4-methyl-; Cedrol
Ethers	n-Butyl ether; 2,5-Dihydroxyacetophenone, bis(trimethylsilyl) ether
Others	3,5-Dimethylpyrazole; 7H-Dibenzo[b,g]carbazole, 7-methyl-; *N*-carbobenzyloxy-l-tyrosyl-l-valine

**Table 5 molecules-25-00471-t005:** The compressive strength and pH value of RP before and after treatment.

Treatment	Compressive Strength (MPa)	Standard Deviation	pH Value
Untreated	68.25	7.46	4.54
Sodium bicarbonate treatment	59.88	5.01	4.95
Ozone treatment	63.71	7.36	4.56

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
