# Peer review of "Comparison on Reduction of VOCs Emissions from Radiata Pine (Pinus Radiata D. Don) between Sodium Bicarbonate and Ozone Treatments"

_molecules, 2020, doi:10.3390/molecules25030471_

Round 1
Reviewer 1 Report
There are at least two issues that have to be corrected.
The first one is connected with the interpretation of GC-MS results which, in my opinion, is not correct. I advise to check available literature data (e.g. L.G. Cool, E. Zavarin, Biochemical Systematics and Ecology 1992, 20, p. 133-144, or A.G. McDonald et. al., Holz als Roh- und Werkstoff, 2002, 60, p. 181-190).
The second one is connected with the presented IR spectra and their interpretation, in my opinion, they are not reliable. Probably adding an internal standard to the studied samples can help to solve this problem. Absorbances of the selected bands divided by an absorbance of an internal standard band should be much more reliable.
A number of mistakes can be found in the reference section.
Author Response
We must thank you for the valuable comments and suggestions, which helped improve our manuscript greatly. Please do forward our heartfelt thanks to the reviewers. Based on the comments we received, careful modifications have been made to the manuscript. All changes were marked in red text. We hope that the revised manuscript answered the questions. Below you will find our point-by-point responses to the comments/ questions:
To Reviewer 1:
The first one is connected with the interpretation of GC-MS results which, in my opinion, is not correct. I advise to check available literature data (e.g. L.G. Cool, E. Zavarin, Biochemical Systematics and Ecology 1992, 20, p. 133-144, or A.G. McDonald et. al., Holz als Roh- und Werkstoff, 2002, 60, p. 181-190).
In lines 215 to 217, we referred to the literature you recommended and revised the interpretation of partial GC-MS results. “Studies had found that pinene compounds in pine were easy to be oxidized, which were usually shown as aromatization [38]. Terpenes reacted quickly with ozone or hydroxyl radicals to produce different organic compounds, including aldehydes and small amounts of ketones [39].” Please check.
References:
[38] Cool, L. G., & Zavarin, E. Terpene variability of mainland Pinus radiata. Biochem. Syst. Ecol. 1992, 20, 133-144.
[39] McDonald, A. G., Dare, P. H., Gifford, J. S., Steward, D., & Riley, S. Assessment of air emissions from industrial kiln drying of Pinus radiata wood. Holz Roh. Werkst. 2002, 60, 181-190.
The second one is connected with the presented IR spectra and their interpretation, in my opinion, they are not reliable. Probably adding an internal standard to the studied samples can help to solve this problem. Absorbances of the selected bands divided by an absorbance of an internal standard band should be much more reliable.
We converted Figure 1 to absorbance as ordinate and added the internal peak. The relatively stable aromatic ether at 1270 cm-1 was selected as the internal control peak, and other control values were obtained as shown in Table 2. In lines 174 to 178, we added a paragraph describing the relative change in the peak value of the stretching vibration absorption of aliphatic ether. “The absorption peaks at 1160 cm-1 and 1109 cm-1 were generated by expansion vibration absorption of aliphatic ether. Table 2 shows that the relative value of fat ether of ozone-treated samples is smaller than that of sodium bicarbonate treated samples. It can be seen that after ozone and sodium bicarbonate treatment, the radiata pine samples will produce aliphatic ether, while sodium bicarbonate treatment produced more.”
A number of mistakes can be found in the reference section.
We have checked the references and corrected the mistakes. Please check.
Sincerely yours,
Ye Qin, Fei Qi, Zhiping Wang, Xianbao Cheng, Botao Li, Anmin Huang, Ru Liu
Jan 6, 2020

Reviewer 2 Report
Molecules
Review of Manuscript Number: molecules-687307
TITLE: Comparison on reduction of VOCs emissions from radiata pine (Pinus radiata D. Don) between sodium bicarbonate and ozone treatments
Authors: Ye Qin, Fei Qi, Zhiping Wang, Xianbao Cheng, Botao Li, Anmin Huang*, Ru Liu
General Comments
This manuscript addresses the application of decomposition method for volatile organic compounds (VOCs) in radiata pine. NaHCO3 or ozone treatments were proposed to actual samples. The proposed method had the possibility of the application for reduction of VOCs technique in the future. I would suggest a minor revision in combination for this manuscript. I annotate the manuscript with several corrections listed below, which I believe will improve the readability of the paper.
Minor comments
Some descriptions are overlapped in Intruduction Section (Line 35: Currently, heat treatment … wood VOCs.; Line 37 – 39: Hyttinen et … of small molecules). I suggest the description of Line 35 should be deleted. SPE is frequently used as solid phase extraction in analytical chemistry. Please replace SPE to SP”M”E (ex. Line 114). Correspondingly, hydroxyl radical is frequently used as “∙ (bullet operation)”OH (ex. Line 208). Line 165 – 168: I could not recognize the difference of transmittance at 1640 cm-1 between NaHCO3 and ozone treatment. Could you please insert some additional explanations and discussions? Some minor grammatical errors were found in the manuscript (Line 253: 110 oC and 200 oC; please correct the degrees to superscript characters).
I hope that my comment is useful for the improvement of the article.
Author Response
We must thank you for the valuable comments and suggestions, which helped improve our manuscript greatly. Please do forward our heartfelt thanks to the reviewers. Based on the comments we received, careful modifications have been made to the manuscript. All changes were marked in red text. We hope that the revised manuscript answered the questions. Below you will find our point-by-point responses to the comments/ questions:
To Reviewer 2:
Some descriptions are overlapped in Intruduction Section (Line 35: Currently, heat treatment … wood VOCs.; Line 37 – 39: Hyttinen et … of small molecules). I suggest the description of Line 35 should be deleted.
We found this problem and deleted the repetitive description in line 35. Please check.
SPE is frequently used as solid phase extraction in analytical chemistry. Please replace SPE to SP”M”E (ex. Line 114). Correspondingly, hydroxyl radical is frequently used as “∙ (bullet operation)”OH (ex. Line 208).
We found this error and changed in lines 114, 116 (SPME), 221, 222, 225, 227, 228 (·OH). Please check.
Line 165 – 168: I could not recognize the difference of transmittance at 1640 cm-1 between NaHCO3 and ozone treatment. Could you please insert some additional explanations and discussions?
We converted Figure 1 to absorbance as ordinate and added the internal peak. The relatively stable aromatic ether at 1270 cm-1 was selected as the internal control peak, and other control values were obtained as shown in Table 2.
At the wavelength of 1640cm-1, the peak area value of the samples treated with sodium bicarbonate and ozone decreased relative to the reference sample and the peak at 1640cm-1 belonged to the skeleton vibration of aromatic ring, suggesting that the content of aromatic hydrocarbons decreased. Meanwhile, the peak at 1640 cm-1 was C=C and C=C will break under ozone oxidation. As can be seen from Table 2, the peak area value of reference sample at 1640 cm-1 wavelength was lower after ozone treatment.
Some minor grammatical errors were found in the manuscript (Line 253: 110 oC and 200 oC; please correct the degrees to superscript characters).
We found this error and changed in lines 264, 265. Please check.
Sincerely yours,
Ye Qin, Fei Qi, Zhiping Wang, Xianbao Cheng, Botao Li, Anmin Huang, Ru Liu
Jan 6, 2020

Round 2
Reviewer 1 Report
The most important issues that should be corrected is connected with the interpretation of GC-MS results. In my opinion, interpretation presented in the manuscript is not correct. Again I advise to check available literature data (e.g. L.G. Cool, E. Zavarin, Biochemical Systematics and Ecology 1992, 20, p. 133-144, or A.G. McDonald et. al., Holz als Roh- und Werkstoff, 2002, 60, p. 181-190). Authors list in Table 3 compounds different from those that are commonly accepted to be present in Pinus radiata wood. Identification of these compounds based only on MS results is not reliable.
Lines 161-164 and 174-175 need to be modified.
Some mistakes can be found in the reference section.
Author Response
We must thank you for the valuable comments and suggestions, which helped improve our manuscript greatly. Please do forward our heartfelt thanks to the reviewers. Based on the comments we received, careful modifications have been made to the manuscript. All changes were marked in red text. We hope that the revised manuscript answered the questions. Below you will find our point-by-point responses to the comments/ questions:
To Reviewer 1:
The most important issues that should be corrected is connected with the interpretation of GC-MS results. In my opinion, interpretation presented in the manuscript is not correct. Again I advise to check available literature data (e.g. L.G. Cool, E. Zavarin, Biochemical Systematics and Ecology 1992, 20, p. 133-144, or A.G. McDonald et. al., Holz als Roh- und Werkstoff, 2002, 60, p. 181-190). Authors list in Table 3 compounds different from those that are commonly accepted to be present in Pinus radiata wood. Identification of these compounds based only on MS results is not reliable.
Different selection and testing methods of wood samples will objectively lead to different results of GC-MS. Compared with the results in the previous studies, the difference in VOCS data is mainly due to the differences in the samples themselves. As a biomass material, wood is an anisotropic material, which could cause the differences.“Studies have shown that wood VOCs mainly originate from their extraction [15].” The content of extract is closely related to the age of the tree and the cutting season of the tree [1]. The samples of radiata pine used in this paper were from New Zealand. The samples of pine radiata used in this paper were cut from New Zealand in autumn. The trees were 15 years old, with an average height of 18.4m and the average diameter at breast height was 20.66cm. All the above reasons will affect the judgment of the main components of VOCs in samples. Considering the above factors, the sapwood samples from different parts of radiata pines were selected in this paper. After the multiple verifications, the main components of the samples were finally determined and listed, which can be considered to be reliable data.
The major VOC compounds in Reference 39 and 40, such as pinene and furfural, are consistent with the major compounds measured in this paper. Part of the VOCs compounds detected are different from those in References 39 and 40 due to the following two reasons.
Differences in sample selection: In this paper, the sapwood samples of radiata pine were tested. In Reference 39, incremental drilling was used to collect samples (including both heartwood and sapwood), and the extracted components of heartwood and sapwood were significantly different, which was one of the reasons for the inconsistency between the experimental results of this paper and Reference 39. In Reference 40, a large number of radiata pine wood blocks were stacked in the industrial kiln and the gas sample was picked up from the kiln. Because the sample gas was the mixture of gases in the kiln, the researchers only selected and characterized the main compounds. This is the reason for the inconsistent between the results in this paper and the literature. . The differences in representational means were tested. The treatment of this paper is described as “The wood powder was placed in a sealed overhead bottle, and placed in a 60 °C water bath pot after using a 100 μm PDMS SPME head to adsorb the volatile organic compounds. The adsorption time was 40 min. The solid phase micro extraction head was then inserted into the gas chromatographic inlet, and the analysis was conducted for 4 min.” The treatment in Reference 39 was “The wood or foliage sample was chopped with a razor blade, placed in a small stoppered Erlenmeyer flask and held at about 35°C for 10 rains to saturate the headspace with vapor. A 200-μl headspace sample was withdrawn with a warm syringe and injected into the GC, which was detected by FID.” The accuracy of this method was the main object of this study. In this paper, SPME was used for radiata pine powder. This method had a low detection limit and can detect more VOC compounds. At the same time, MS has higher sensitivity and lower detection limit than FID, which can characterize more specific VOC compounds in samples. This is one of the reasons that the experimental results of this paper are inconsistent with those of References 39 and 40.The main purpose of this paper is to investigate whether the oxidation and alkali treatment are effective in removing VOCs from radioactive pine. The substances listed in Table 3 in this paper were partly produced after the treatment, which were not all originally contained in radiata pine samples. In this paper, the FTIR results, microscopic images, mechanical test results and GC-MS results correspond to each other, and there is no single finding only depended on the examination of GC-MS.
Ref:
[1] Hillis, W. E. "Distribution, properties and formation of some wood extractives." Wood Sci. technol. 1971,5, 272-289.
[15] Liu R, Wang C, Huang A, Lv B. Characterization of odors of wood by gas chromatography-olfactometry with removal of extractives as attempt to control indoor air quality. Molecules 2018,23, 203.
[39] Cool, L. G., & Zavarin, E. Terpene variability of mainland Pinus radiata. Biochem. Syst. Ecol. 1992, 20, 133-144.
[40] McDonald, A. G., Dare, P. H., Gifford, J. S., Steward, D., & Riley, S. Assessment of air emissions from industrial kiln drying of Pinus radiata wood. Holz Roh. Werkst. 2002, 60, 181-190.
Lines 161-164 and 174-175 need to be modified.
Lines 165-171, the pure comparison of the peak area value were deleted and the reason for group transformation was inferred according to the literature. “Both of these treatments will cause C=C to be destroyed. The Alkali treatment can remove lignin in wood powder, expose more cellulose and increase the concentration of hydroxyl on the surface of cellulose [38]. The Ozone treatment can oxidize lignin and lead to hemicellulose degradation.”
Lines 179-181, re-conjecture the cause of the transformation of peaks 1160 and 1109. “Both of these reactions were accompanied by the formation of small molecules, so there was an increase in the group at peaks 1160 and 1109.”
Ref:
[38] Chang, W. P., Kim, K. J., & Gupta, R. K. Ultrasound-assisted surface-modification of wood particulates for improved wood/plastic composites. Compos. Interface. 2009, 16, 687-709.
Some mistakes can be found in the reference section.
We double checked References and made the changes accordingly.
Sincerely yours,
Ye Qin, Fei Qi, Zhiping Wang, Xianbao Cheng, Botao Li, Anmin Huang, Ru Liu
Jan 16, 2020
